# Gaming Disorder and Psycho-Emotional Wellbeing among Male University Students and Other Young Adults in Israel

**DOI:** 10.3390/ijerph192315946

**Published:** 2022-11-30

**Authors:** Richard Isralowitz, Shai-li Romem Porat, Yuval Zolotov, Mor Yehudai, Adi Dagan, Alexander Reznik

**Affiliations:** Regional Alcohol and Drug Abuse Research Center, Ben Gurion University of the Negev, Beer Sheva 84105, Israel

**Keywords:** internet gaming, gaming disorder, university students, male adults, psycho-emotional wellbeing, substance use, eating behavior

## Abstract

The aim of this cross-sectional study was to examine the impact of gaming and gaming disorder on the wellbeing of Israeli male university students and other adults. Gaming disorder (i.e., persistent, and recurrent gaming activity associated with a lack of control that may be clinically diagnosed) was determined using the Internet Gaming Disorder Scale–Short-Form (IGDS9-SF). Survey participants were recruited from gaming associations, clubs and the gaming community using Facebook. Data were collected in June 2022. A total of 526 males completed the survey (30.9% university students and 69.1% other young adults). Various statistical methods of analysis including regression were used for this study. Significant study group differences revealed university students with more indications of gaming disorder, more burnout, less loneliness, more stimulant (i.e., Ritalin) use, a greater consumption of salt- and/or sugar-loaded foods and lower economic wellbeing. The levels of resilience (i.e., the ability to recover from stress), substance use (e.g., tobacco and alcohol) and weight gain were similar for the two groups. Regression analysis showed gaming disorder as a key predictor of burnout, economic wellbeing and resilience. This study examined only male gamers because of the small number of female respondents. However, additional research is needed about female internet gamers, including their possible exposure to online harassment and sexual degradation. Additionally, additional research should be considered to verify the present study’s findings about gamers based on demographic factors and gaming disorder levels. Prevention and treatment intervention measures, including those that can be made available on campus, should be thought about by university administration personnel and student association leaders in consultation with professionals who are experienced in reducing gaming disorder and other harmful behaviors among students.

## 1. Introduction

Gaming is a behavior commonly played either through a computer, a gaming console or any other device, either online or offline [1]. Interest in video games has increased, with approximately 3 billion people worldwide playing video games and an industry valued at about USD 300 billion, which is attributed to the mobile nature of the activity and pandemic confinement [2]. Frequent internet gamers tend to be youth and young adults. However, older adults, including those over 65 years of age, engage in the activity [3].

Video gaming has impacts that justify evidenced-based research for informed decision making among policymakers, health professionals, educators, and others [4]. For example, persistent and recurrent gaming can lead to gaming disorder (GD), which may be associated with significant impairments of daily, work and/or education activities. The American Psychiatric Association has suggested that this condition is a tentative psychiatric disorder requiring further study (IGD) [5,6]. Research on GD and IGD has evolved considerably in recent years, with meaningful findings about psychological distress, stress, depression, and anxiety [7,8,9,10,11,12]; internet gaming disorder [13,14,15]; and the deterioration of psychosocial, physical, and social conditions [7,15,16,17]. From a beneficial perspective, gaming may promote cognition (i.e., mental action acquiring knowledge and understanding), eye–hand coordination [18,19], memory and stress reduction, among other conditions [19,20,21,22,23].

The outbreak of the COVID-19 pandemic, beginning in November 2019, was followed by mitigation measures including lockdowns and social distancing, which, in turn, have been associated with health and mental health conditions [24]. Under these restrictions, the popularity of video gaming increased as a leisure-time activity and coping strategy against distress [15,25,26]. This increase was most prevalent among young adults [27], and it is plausible that gaming had a positive effect on their wellbeing as players by providing an enjoyable means of maintaining social contact and an opportunity to escape from negative pandemic-related effects [28].

### The Current Study

The objective of this exploratory study was to better understand the association of gaming and gaming disorder with the wellbeing of young adults in Israel. This study was not intended to generate conclusive results but rather provide usable information to guide future research that may be used for policy, prevention, and treatment intervention purposes. A recent but limited study of Israeli university students evidenced that two-thirds of the respondents engaged in weekly gaming behavior (67.7%), and among those gamers, more than 20% spent at least 10 h a week on such activity [29]. Another study showed that gaming activity is dominated by males, who commonly engage in online sex-oriented conversations and harassment of female gamers [30].

## 2. Methods

Established in 1996, the Ben Gurion University of the Negev (Israel), Regional Alcohol and Drug Abuse Research (RADAR) Center has received US National Institute on Drug Abuse recognition for its collaborative efforts. For this study, the RADAR Center contacted computer gaming association and club members as well as the gaming community using Facebook to inform about the online survey and its aim to study gaming and gamer wellbeing. University ethics approval was issued to ensure that appropriate research measures were taken to protect the confidentiality and rights of the survey participants involved. Participants were advised about the survey’s compliance with all ethical standards, the confidentiality of their responses and that their responses constituted consent to participate. Data were collected in June 2022 from the gamers who voluntarily participated.

The Qualtrics software platform was used for data collection from respondents to the nine-item Internet Gaming Disorder Scale–Short-Form (IGDS9-SF) [1]. This was the first brief standardized psychometric tool to assess IGD according to the criteria suggested by the American Psychiatric Association in the 5th edition of the Diagnostic and Statistical Manual of Mental Disorders. The levels of agreement with IGDS9-SF statements are evaluated by a 5-point Likert scale from 1 (Never) to 5 (Very Often). Higher total scores correspond with a higher level of internet gaming disorder. Respondents with and without gaming disorder were determined based on their responses. Those endorsing at least five criteria out of the nine were categorized as evidencing gaming disorder [31]. Gaming disorder was categorized into three groups: low (*n* = 126), medium (*n* = 258) and high (*n* = 142).

Other instruments used for the survey data collection were the ten-item Short Burnout Measure (SBM) [32] to assess levels of physical, emotional and mental exhaustion [32]; the six-item Brief Resilience Scale (BRS) to determine a person’s ability to bounce back or recover from stress [33]; the six-item De Jong Gierveld Loneliness Scale (SLS) to gauge overall emotional and social loneliness [34]; and the ten-item Economic Well-Being Scale (EWBS) based on quality of life [35]. Additionally, survey participants were asked questions about their psycho-emotional wellbeing, substance use, eating behavior of unhealthy foods (i.e., those with a high level of salt and/or sugar) and weight gain. The data collection instruments used to assess substance use and eating behavior were adapted from the RADAR Center regional and cross-national research supported by the United States Agency for International Development–Middle East Regional Cooperation Program. During the last 20 years, the Center’s Substance Use Survey Instrument (SUSI) has been modified and updated to address shifting patterns of substance use and behavior among at-risk populations [36,37].

The data collection instruments used for this research were translated into Hebrew and back into English by native speakers of both languages to ensure that the content and vocabulary were appropriate for the gamers surveyed. The survey instruments used showed a good internal consistency, as evidenced by the following Cronbach’s alpha scores: IGDS9-SF = 0.804; SBM = 0.907; BRS = 0.773; SLS = 0.704; and EWBS = 0.913. All statistical analyses were conducted using SPSS, version 25. Stepwise multiple regression, Pearson’s chi-squared test for dichotomous variables, the Mann-Whitney test, a t test and one and two-way ANOVA for continuous variables were used for the data analysis.

A total of 526 young adult males completed the online survey: 30.8% (*n* = 162) university students and 69.1% (*n* = 364) non-students. The mean age of the respondents was 28.5 years (SD = 6.8); students were younger than non-students: 24.9 (SD = 3.5) and 30.0 (SD = 7.4), respectively (t_520_ = 8.347; *p* < 0.001).

## 3. Results

The Internet Gaming Disorder Scale–Short-Form (IGDS9-SF) respondent average score was 19.1 (SD = 5.9), and its distribution was close to normal (skewness was 0.685 and kurtosis was 0.331). The mean IGDS9-SF scores were 20.0 (SD = 6.2) and 18.7 (SD = 5.7) for the students and non-students, respectively (t_524_ = 2.201; *p* = 0.028). No significant gaming disorder value differences were found when respondents were compared based on religiosity (i.e., secular and non-secular) (t_524_ = 1.907; *p* = 0.057) and marital (i.e., married, partnered or single) statuses (t_524_ = 1.964; *p* = 0.050).

The mean SBM (burnout) score for all respondents was 24.2 (SD = 9.2), and university students reported a significantly higher level of burnout than non-students: 26.5 (SD = 9.0) and 23.1 (SD = 9.2), respectively (t_444_ = 3.597; *p* < 0.001). No significant burnout score differences were found among the respondents based on religiosity (t_444_ = 0.023; *p* = 0.892) and marital statuses (t_444_ = 0.830; *p* = 0.407). However, a statistically significant association between burnout and gaming disorder level was found. High burnout was associated with high gaming disorder (F_2429_ = 49.297; *p* < 0.001).

Responses to the De Jong Gierveld Loneliness Scale (SLS) evidenced a median score of 3.0, with an interquartile range (IQR) of 3.0. University students reported a lower level of loneliness than other adults (U = 16,910.00; *p* = 0.025). No significant differences were found for loneliness level based on religiosity (U = 10,516.50; *p* = 0.551) and marital statuses (U = 21229.000; *p* = 0.193). A high level of loneliness was associated with a high level of gaming disorder (F_2411_ = 12.084; *p* < 0.001).

The average Brief Resilience Scale (BRS) value for all respondents was 20.7 (SD = 4.2). No significant differences were found for student and non-student resilience scores (t_458_ = 0.423; *p* = 0.672) and marital status (t_458_ = 1.021; *p* = 0.308). However, religious students reported a higher resilience level than secular students (t_458_ = 2.010; *p* = 0.045). A statistically significant association between resilience and gaming disorder was found. High gaming disorder was associated with low resilience (F_2443_ = 11.987; *p* < 0.001).

The rate of last-month substance use was similar among the students (81.3%) and other adults (82.2%) (χ^2^(1) = 0.056, *p* = 0.813). However, the last-month use of Ritalin (i.e., a synthetic stimulant used to promote mental activity) was more prevalent among students than non-students (14.0% vs. 5.9%; χ^2^(1) = 8.488, *p* = 0.004). Table 1 and Table 2 present the prevalence of substance use and psycho-emotional wellbeing cross-tabulated with the three gaming disorder levels.

Regarding eating behavior, 26.9% of the gaming respondents reported that they consumed more food with high levels of salt (e.g., chips, sausage, and cheese) and/or sugar (e.g., sweets and chocolate). This behavior was more prevalent among university students than other adults (34.3% vs. 23.5%; χ^2^(1) = 5.656, *p* = 0.017). Additionally, 47.0% of the respondents reported a weight increase during the pandemic, a level similar among students (48.6%) and non-students (46.3%) (χ^2^(1) = 0.206, *p* = 0.652). Those who reported eating unhealthy food and weight gain had higher IGDS9-SF gaming disorder scores: t_434_ = 6.222, *p* < 0.001 and t_433_ = 2.590, *p* = 0.010, respectively.

The average economic wellbeing value (EWBS) for all respondents was 36.0 (SD = 9.4). The EWBS scores were significantly lower among students (32.6; SD = 9.3) than non-students (37.7; SD = 9.0) (t_430_ = 5.460; *p* < 0.001). No significant differences were found for economic wellbeing based on respondent religiosity (t_430_ = 0.798; *p* = 0.425) and marital statuses (t_430_ = 1.097; *p* = 0.237). Economic wellbeing scores were associated with gaming disorder level. High gaming disorder was associated with low economic wellbeing (F_2,415_ = 10.836; *p* < 0.001).

Stepwise multiple regression evidenced that gaming behavior score, and student status are key predictors of burnout and economic wellbeing. The values of the explained variance of these dependent variables (adjusted R^2^) were 0.256 (25.6%) and 0.107 (10.7%), respectively. Additional analysis showed gaming disorder and religiosity to be significant predictors of resilience. The value of the explained variance (adjusted R^2^) for resilience was 0.069 (6.9%). No other factors (e.g., age and marital status) were found to be significant predictors.

## 4. Discussion and Conclusions

This study aimed to assess gaming and gaming disorder among male university students and other young adults in Israel and to examine its association with different domains of psycho-emotional wellbeing. Among the 526 participants who completed the survey, most respondents (76%) evidenced medium or high levels of gaming disorder. Although the categorization and extent of gaming disorder were driven by the unique characteristics of our sample, we believe that this distribution is a concern warranting further study over time and across locations for possible policy and prevention intervention purposes.

High levels of gaming disorder were found to be significantly associated with the deterioration of mental and psycho-emotional conditions. These findings are consistent with those of other studies that point to the association of gaming disorder with the deterioration of various mental, psychological, and psycho-emotional conditions [7,8,9,10,11,12]. University students, compared to other adults, scored higher in terms of internet gaming disorder measurement. Furthermore, they (i.e., the university students) reported more burnout and stimulant use (i.e., Ritalin) as well as less economic wellbeing and healthy eating behavior. Last-month substance use, particularly alcohol, was reported to be high by all respondents (75.3%). In terms of loneliness, university students reported a lower level than the non-students surveyed. This finding suggests that being a member of an academic institution and community may provide positive emotional and social support that mitigates loneliness.

Among the study limitations, this cross-sectional survey may not be representative of the Israeli gaming population, and the generalization of the findings for gamers in the country and elsewhere should be made with caution. Furthermore, this study examined only male gamers because of the small number of female respondents. Additional research is needed regarding internet gaming among females and their possible exposure to online harassment and sexual degradation based on gender status. Another shortcoming of the present study is that eating behavior among gamers was looked at only in terms of unhealthy food intake and weight gain. Research needs to examine the impact of gaming behavior on missed meals and weight loss as well.

In conclusion, this study shows that university students are at risk of gaming disorder, which may affect mental and psycho-emotional wellbeing, as well as academic performance. Prevention and treatment intervention measures, including those that can be made available on campus, should be considered by university administration personnel and student association leaders in consultation with professionals who are experienced in reducing gaming disorder and other harmful behaviors among students.

## Figures and Tables

**Table 1 ijerph-19-15946-t001:** Last-month substance use by gaming disorder level.

	Gaming Disorder Levels			
	Low (*n* = 126)	Medium (*n* = 258)	High (*n* = 142)	χ^2^ Test *p* Value
Tobacco, % (n)	26.4 (28)	25.9 (60)	20.8 (26)	0.507
Alcohol, % (n)	75.5 (80)	79.0 (184)	69.6 (87)	0.144
Cannabis, % (n)	29.5 (31)	24.3 (56)	25.2 (31)	0.595
Ritalin, % (n)	11.5 (12)	6.2 (14)	9.0 (11)	0.234
Pain relievers, % (n)	6.7 (7)	9.6 (22)	12.3 (15)	0.359
Sedatives, % (n)	3.8 (4)	3.6 (8)	9.8 (12)	0.033
Any substance use, % (n)	80.6 (83)	84.1 (185)	78.8 (93)	0.452
Binge drinking, % (n)	2.8 (3)	1.3 (3)	5.6 (7)	0.068

**Table 2 ijerph-19-15946-t002:** Last-month psycho-emotional wellbeing by gaming disorder level.

	Gaming Disorder Level			
	Low (*n* = 126)	Medium (*n* = 258)	High (*n* = 142)	χ^2^ Test *p* Value
Depressed, % (n)	6.6 (7)	14.7 (34)	27.2 (34)	<0.001
Exhausted, % (n)	17.1 (18)	19.5 (45)	47.2 (59)	<0.001
Lonely, % (n)	5.7 (6)	13.1 (30)	22.4 (28)	<0.001
Nervous, % (n)	13.3 (14)	21.2 (49)	41.3 (52)	<0.001
Angry, % (n)	6.7 (7)	13.0 (30)	35.7 (45)	<0.001
General deterioration of psycho-emotional wellbeing, % (n)	28.6 (30)	34.6 (79)	60.2 (74)	<0.001

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
