# Peer review of "Gaming Disorder and Psycho-Emotional Wellbeing among Male University Students and Other Young Adults in Israel"

_ijerph, 2022, doi:10.3390/ijerph192315946_

Round 1

Reviewer 1 Report

An interesting study that may benefit from some additional work and revisions to be ready for publication:

1. Have the entire manuscript edited for language by a professional editor. See for example the first 2 lines of the Abstract ("devise" instead of device, etc.). 

2. In the abstract there is too much background and too little methodological and results related info. Shrten the "exposition"  - start with the aim of the study. Add a short decription of what gaming disorder is (is it a clinical psychological diagnosis? what does it describe?) and how  was it assessed, in addition to the outcome variables. Also avoid claiming that gaming leads to psychological challnges - it may as well be the other way around - people who suffer from distress turn to gaming as "pain relief". Keep your language clear re the correlational nature of the study. 

3. The introduction needs to establish pretty early on in the text "what's the problem with gaming" - and what is considered excessive gaming or a disorder. Gaming per se was always there - with and without computers. The authors MUST clarify what seems to be the problem or the risk in gaming behavior and the "slippery slope" it often presents to "gamers". 

4. The intro must be further developed. As it sits - it is not clear what gaming is, whether it is a disorder (as hinted in the abstract) or not and what behaviors can present certain dangers to well being that are included in "gaming". The  benefits and risks associated with gaming need to be elaborated and a question or problem presented before presenting the rationale for the current study. 

5. The rationale for the study must be further developed. A lack of information on gaming cannot be the reason for this study - but rather a reason to conduct a national level survey of gaming behaviors - not sure this is what the study is about. Whay information can be gleaned from the current study that will shed new light on our understanding of the risks, and perhaps other outcomes of gaming? and when gaming behavior tunrs into a disorder?

6. The methods section is lacking and disorganized. Re-structure it as follows:

Start with the settings - where was the study conducted ? which populations were used as the frame of sampling?

Then describe your sample: sampling method, and then sample demographics.

Measures used:

For each varibale included in your study specify - the measure used, describe it, the source it is taken from and preliminary info re its reliability and validity. 

Procedure: how potential particiapnts were recruited, what were they told? how were their rights preserved? who approved the study?

7. In the results - please explain the rationale for determining "gaming disorder" on what basis do the authors determine that a certain amount of time spent gaming reflects a disorder?

In the statistical analyses: your sample is large enough to allow association based analyses. Distribution comparisons as condected are very inaccurate and convoluted manner of testing for associations. I would stringly suggest using stepwise multiple linear regressions using demographics as the first step, then add the gaming score in the second to "predict" each of the outcome scores - this will provide cleaner assessments of the independent variables contribution to the variance in each of the outcome measures. 

8. The discussion is too short and dull. I will refer to my comments above in challenging the idea of "gaming disoprder" found in the sample? since we do not know how the authors defined the disorder - it is impossible to understand the meaning of these statements. 

The comes the "so what?" question: so we found the associations between extent opf gaming hours and some indicators of psychological distress. SO? discuss the association, explain what may account tfor it but be careful - the correlational nature of the stay design basically can mean that gaming leads to deteriorated psychological function but it can also mean the other way around - maybe more burnt out, depressed individuals turn to gaming?

Include a section dealing with the study limitations and what can be done in furute studies to address them. 

Author Response

Response to Reviewer 1 Comments

Point 1:  Have the entire manuscript edited

Response 1:  The manuscript has been reviewed and revised by a professional editor. Location within the manuscript - throughout.

Point 2: Abstract revision

Response 2: Abstract has been revised considerably, in accordance with reviewer's suggestions. Specifically, the background has been reduced, and the abstract starts now with the aim of the study.  We added details about gaming disorder, methodological information and results.  Finally, we clarified the language to reflect the correlational nature of our study.  Location in the manuscript - Abstract p. 2 

Point 3. Introduction 

Response 3:  The introduction has been revised to address the impact of gaming in terms of negative and positive factors.  We believe our revisions currently clarify what might be the problems and risks associated with excessive gaming. Location in the manuscript - Page 3, paragraph 2

Point 4:  Introduction must be further developed 

Response 4: We revised the introduction to reflect the comments made by the reviewer.  We believe it currently conveys more clearly which benefits and risks are associated with gaming. Location in the manuscript - Page 3, paragraphs 2-3

Point 5: Study rationale

Response 5: The rationale has been revised; mention is made that this is a cross-sectional paper. Also, mention about a lack of information has been removed - Location in the manuscript - Page 3, paragraph 4

Point 6: Methods section

Response 6: The methods section has been revised and edited in accordance with the reviewer's comments. Specifically, we provide details about the sample, the recruitment and the demographics, as well as about participants' rights and ethical approval.  We additionally include relevant information about each variable included in our study (measurement used, source, and internal reliability and validity.  Location in the manuscript - Page 4, paragraphs 3-4 and Page 5, paragraphs 1-3. 

Point 7:  In the results - please explain the rationale for determining gaming disorder

Response 7: In light of this comment, we added details about gaming disorder, the analysis, section has been reworded and stepwise regression information added.  As suggested by the reviewer, we added stepwise linear regression models with demographics and gaming disorder score as potential predictors for resilience, burnout and economic well-being.  Location in the manuscript - Throughout the results section (pages 5-8) and specifically Page 8 -paragraph 1 

Point 8: Discussion 

Response 8: The discussion section has been edited; limitations added including the need for further study, and recommendations added that are relevant for prevention and intervention.  Location in the manuscript - Discussion and conclusion section p. 11

Reviewer 2 Report

Thank you for the opportunity to review the manuscript titled "Gaming Disorder and Psycho-emotional Wellbeing among Male University Students and Other Young Adults in Israel" for consideration of publication.  The researchers investigated how increased time spent on computer and internet gaming among college students (as compared to non-college controls) negatively affects variables associated with well-being.

The abstract is succinct and adequately reports the purpose, methods, and results of this research study.  Introduction is adequate for describing the background of internet gaming disorder and lays a foundation for why this current investigation is warranted.  Methods section adequately describes participants, how they were recruited, how data was collected, and how the data was analyzed.  Results and statistical analysis appear appropriate and provide meaningful results.  Discussion follows well from data analysis, and conclusions appear to be valid given the results.

The authors do a good job presenting their research, and this study is well-grounded in methodology and data analysis.  I believe a real strength of this research is that the authors chose to use a non-student comparison group in addition to grouping college students based on the amount of internet gaming in which they engaged.  This procedure provides both internal comparisons amongst college students as well as external comparisons, bolstering the authors' ability to draw conclusions from their data.

I have no major concerns about this paper.  My only suggestion would be to provide a bit more detail about "gaming disorder" in the introduction, as it is likely that this is a diagnosis that many readers will have limited familiarity.

Author Response

Point 1: More detail about gaming disorder in the introduction 

Response 1: The introduction has been revised to provide more detail about gaming disorder.  Location in manuscript - Page 3, paragraph 2